# Associations between Mental Health and COVID-19 Status among 18- and 19-Year-Old Adolescents: A Multi-Country Study

Morenike Oluwatoyin Folayan [1,2,3,*], Roberto Ariel Abeldaño Zuñiga [1,4], Mir Faeq Ali Quadri [1,5], Joanne Lusher [1,6], Balgis Gaffar [1,7], Passent Ellakany [1,8], Annie L. Nguyen [1,9] and Maha El Tantawi [1,10]

1 MEHEWE Study Group, Obafemi Awolowo University, Ile-Ife 220282, Nigeria
2 Department of Child Dental Health, Obafemi Awolowo University, Ile-Ife 220282, Nigeria
3 Nigeria Institute of Medical Research, Yaba, Lagos 100001, Nigeria
4 Postgraduate Department, University of Sierra Sur, Oaxaco 70805, Mexico
5 Department of Oral Health Sciences, School of Dentistry, University of Washington, Washington, DC 98195, USA
6 Provost's Group, Regent's University, London NW1 4NS, UK
7 Department of Preventive Dental Sciences, College of Dentistry, Imam Abdulrahman Bin Faisal University, Dammam 31441, Saudi Arabia
8 Department of Substitutive Dental Sciences, College of Dentistry, Imam Abdulrahman Bin Faisal University, Dammam 31441, Saudi Arabia
9 Department of Family Medicine, Keck School of Medicine, University of Southern California, Los Angeles, CA 98105, USA
10 Department of Pediatric Dentistry and Dental Public Health, Faculty of Dentistry, Alexandria University, Alexandria 21544, Egypt
* Correspondence: toyinukpong@yahoo.co.uk

**Abstract:** The aim of this study was to describe the mental health status of 18- and 19-year-old adolescents who were infected or affected by COVID-19 during the first wave of the COVID-19 pandemic. This was a secondary analysis of a dataset collected from 152 countries between July and December 2020. Dependent variables were anxiety, depression, and post-traumatic stress symptoms. The independent variable was COVID-19 status (tested positive for COVID-19, had COVID-19 symptoms but did not test, had a close friend who tested positive for COVID-19, knew someone who died from COVID-19). Three multivariable logistic regression analyses were conducted to determine the associations between the dependent and independent variables while adjusting for confounding variables (sex—male, female, and country income level). Data of 547 participants were extracted, and 98 (17.9%) had experienced depression, 130 (23.8%) had experienced anxiety, and 219 (40.0%) had experienced post-traumatic stress symptoms. Knowing someone who died from COVID-19 was associated with significantly lower odds of having post-traumatic stress symptoms (AOR: 0.608). Having COVID-19 symptoms but not getting tested was associated with significantly higher odds of having anxiety symptoms (AOR: 2.473). Results indicate diverse mental health responses among adolescents aged 18–19-years old as a sequela of COVID-19. This needs to be studied further.

**Keywords:** COVID-19; mental health; adolescent; stress; anxiety; post-traumatic stress

## 1. Introduction

COVID-19 severity varies by sex, socio-economic context, and life circumstances [1], and adolescents are generally at lower risk for severe COVID-19 compared to other age groups [2,3]. However, the COVID-19 pandemic and associated lockdowns took a significant toll on the mental wellbeing of adolescents [4]. Substantial increases in emotional symptoms and hyperactivity and inattention among adolescents experiencing psychosocial problems were seen over the pandemic period [5,6]. Restrictions in normal school and peer activities [7,8] have led to worsening mental health outcomes, including rising rates of

anxiety and depression [9–11]. Restrictions on movement and engagement in public and communal spaces, limited opportunities for physical activity, and interrupted sleep patterns also contributed to poor mental health outcomes [12,13]. Differences among adolescents in their ability to access social support [14], maintain resilience [15], and use positive coping strategies to respond to COVID-19 pandemic-related stress [16] impact the development of mental health issues [17].

Some studies have considered the pandemic's impact on the mental health of adolescents who identify with specific groups, such as LGBT youth [18] and youth living with HIV [19]. It is vital that we understand how the pandemic may have affected the mental health of adolescents within a homogenous cluster. Adolescents can be divided into younger (10–13 years old) and older (14–19 years old) categories based on their stage of physical and emotional development. Older adolescents can be further divided into 14–17 years old and 18–19 years old based on their civic and legal rights [20]. In most countries, adolescents aged 18 and 19 years are legally independent and presumed to have the competency to make decisions about their health care needs [21]. This period of emerging adulthood [22] is associated with reaching physical, cognitive, and emotional development and assuming responsibilities associated with adulthood [23]. Identifying risk factors for mental health challenges in this subpopulation of adolescents, who are independent of parental care, will assist in planning mental health care services for this population during a health crisis.

Earlier studies conducted in adult populations or in populations that included adolescent 18–19-year-olds showed that having COVID-19 is associated with higher risk of depression, anxiety [24], and post-traumatic stress disorder [25]. Having a friend or family member who tested positive for COVID-19 or knowing someone who died from COVID-19 were associated with higher risk of anxiety [26] and post-traumatic stress disorder [27]. Moreover, having COVID-19 symptoms without testing was associated with depression [27]. The aim of the current study was therefore to extend our understanding of the impact of COVID-19 on mental health by describing the mental health status of 18- and 19-year-old adolescents infected or affected by COVID-19 during the first wave of the COVID-19 pandemic. Specifically, it explored associations between depression, anxiety, and post-traumatic stress symptoms and COVID-19 status among adolescents aged 18 or 19. It was hypothesised that COVID-19 status would be positively associated with the experience of depression, anxiety, and post-traumatic stress symptoms among 18- and 19-year-old adolescents.

## 2. Materials and Methods

This was a secondary analysis of a dataset generated through a large cross-sectional multicountry study that determined the impact of COVID-19 on the mental health and wellness of 21,206 adults 18 years and above. The data was generated from 152 countries between July and December 2020 using an online questionnaire validated for global use [28]. The overall content validity index of the questionnaire was 0.83 [28]. The data of adolescents 18 and 19 years old were extracted for this study.

Ethical approval for this study was obtained from the Human Research Ethics Committee at the Institute of Public Health of the Obafemi Awolowo University Ile-Ife, Nigeria (HREC No: IPHOAU/12/1557). Additional ethical approvals were obtained from India (D-1791-uz and D-1790-uz), Saudi Arabia (CODJU-2006F), Brazil (CAAE N° 38423820.2.0000.0010), and the United Kingdom (13283/10570). Study participants provided consent before participating in the online survey.

*Sample size*: The extracted data of 547 adolescents were considered statistically adequate as there was a minimum of 10 participants with complete responses for each of the independent variables applied in this study. This enabled us to perform regression analyses with a minimum probability level of 0.05 [29].

*Participant recruitment*: Participants were recruited for the primary study using an online survey tool (Survey Monkey®). The link to the survey questionnaire was publicly

shared on social media (Facebook, Twitter, and Instagram), email lists, and WhatsApp groups. Respondents were asked to share the link further with their networks. Respondents had to be 18 years or older, understand the survey language (Arabic, English, French, Portuguese, or Spanish), and have an electronic device and internet connection to be able to access the survey. Details of the study participants and recruitment process have been previously published [28,30].

*Study procedure*: The survey was preceded by a brief introduction explaining the objectives of the study. Participants were assured of confidentiality and the voluntary nature of the study. Only participants who indicated consent by checking a box were able to proceed with the survey. The questionnaire took, on average, 11 min to complete. Survey responses completed below 7 min (the lower limit of the time range to answer the questionnaire during the piloting of the instrument) were excluded from data analysis. Furthermore, incomplete data were removed [31,32]. IP address restrictions was placed thereby limiting each participant to completing only a single questionnaire on a device. Participants could, however, edit their answers freely until they chose to submit. Full details of the methodology are reported elsewhere [28,30].

### 2.1. Dependent Variables

*Anxiety and depression*: Respondents were asked to indicate if they experienced anxiety or depression during the pandemic by checking a response box. Responses were dichotomized into "yes" (if the response was selected) and "no" (if the response was not selected). The questions were adopted from the Pandemic Stress Index [33], which uses a select-all-that-applies format for endorsement of symptoms. The content validity index for this section of the study tool was 0.90 [28].

*Post-traumatic stress symptoms*: The post-traumatic stress disorder checklist for civilians was used to measure the level of post-traumatic stress symptoms (PTSS) that respondents had. The checklist is a 17-item, self-report questionnaire that prompts respondents to measure the level of stress that they have in response to a stressful life experience (in this case the COVID-19 pandemic) over the past month [34]. A 5-point scale was used to rate responses (1,—not at all to 5—extremely) with possible scores ranging from 17 to 85, with higher scores indicating greater risk. The cutoff of 28 was used to categorize responses to "no PTSS" (17–27) vs. "PTSS present" (28–85) [35]. The instrument has good internal consistency, and discriminant validity [34,36–39]. For this study, the Cronbach alpha score was 0.961.

### 2.2. Independent Factors

*COVID-19 status*: Respondents were asked if they tested positive for COVID-19, had COVID-19 symptoms but did not test, had a close friend who tested positive for COVID-19, or knew someone who died from COVID-19. Response choices for each item were "yes" or "no".

### 2.3. Confounding Factors

*Sociodemographic variables*: The confounding sociodemographic variables were sex at birth (male, female) and country income level. Information about country income level was obtained from publicly available data from the World Bank [40]. Low-income countries (LIC) included countries with a gross national income (GNI) per capita of ≤1035 USD in 2019, lower middle-income countries (LMIC) had a GNI between 1036 and 4045 USD, upper middle-income countries (UMIC) had a GNI between 4046 and 12,535 USD, and high-income countries (HIC) had a GNI ≥ 12,536 USD.

### 2.4. Data Analysis

Raw data were downloaded, cleaned, and imported to SPSS version 23.0 (IBM SPSS Statistics for Windows, Version 23.0. Armonk, NY: IBM Corp.) for analysis. Inferential analyses were conducted by developing three multivariable logistic regression models, one

for each dependent variable, to determine the associations between the dependent and independent variables while adjusting for confounding variables. Adjusted odds ratios (AOR) for the multivariable logistic regression models and 95% confidence intervals (CI) were calculated. Statistical significance was set at <0.05.

## 3. Results

Table 1 shows that out of the 547 participants, 353 (64.5%) were 19 years old, 400 (73.1%) were females, 98 (17.9%) had experienced depression, 130 (23.8%) had experienced anxiety, and 219 (40.0%) had experienced post-traumatic stress symptoms. In addition, 33 (6.0%) had SARS-CoV-2 infection, 219 (40.0%) knew of someone who had died from COVID-19, 178 (32.5%) had a close friend or family member who tested positive for SARS-CoV-2, and 64 (11.7%) had COVID-19 symptoms and did not get tested.

Having a close friend and family members who tested positive for COVID-19 was not significantly associated with experiencing depression, anxiety, or having post-traumatic stress symptoms. Knowing someone who died from COVID-19 was associated with significantly lower odds of experiencing post-traumatic stress symptoms (AOR: 0.608; 95% CI: 0.407–0.908; $p = 0.015$). Having COVID-19 symptoms but not getting tested was associated with significantly higher odds of experiencing anxiety symptoms (AOR: 2.473; 95% CI: 1.286–4.755; $p = 0.007$).

**Table 1.** Multivariate logistic regression on the association between COVID-19 status and depressive symptoms, anxiety symptoms, post-traumatic stress symptoms among 18- to 19-year-old adolescents (N = 547).

| Variables | Total N = 547 n (%) | Experienced Depression | | | Experienced Anxiety | | | Post-Traumatic Stress Symptoms | | |
|---|---|---|---|---|---|---|---|---|---|---|
| | | Yes n (%) 98 (17.9) | No n (%) 449 (82.1) | AOR 95% CI *p* Value | Yes n (%) 130 (23.8) | No n (%) 417 (76.2) | AOR 95% CI *p* Value | Yes n (%) 219 (40.0) | No n (%) 328 (60.0) | AOR 95% CI *p* Value |
| **Age in years** | | | | 0.682; 0.415–1.123; *p* = 0.133 | | | 0.935; 0.601–1.456; *p* = 0.767 | | | 0.980; 0.677–1.418; *p* = 0.913 |
| 18 | 194 (35.5) | 28 (14.4) | 166 (85.6) | | 43 (22.2) | 151 (77.8) | | 78 (40.2) | 116 (59.8) | |
| 19 | 353 (64.5) | 70 (19.8) | 283 (80.2) | 1.000 | 87 (24.6) | 266 (75.4) | 1.000 | 141 (39.9) | 212 (60.1) | 1.000 |
| **Sex at birth** | | | | | | | | | | |
| Male | 147 (26.9) | 28 (17.5) | 119 (81.0) | 1.000 | 35 (23.8) | 112 (76.2) | 1.000 | 46 (31.3) | 101 (68.7) | 1.000 |
| Female | 400 (73.1) | 70 (19.0) | 330 (82.5) | 0.815; 0.490–1.353; *p* = 0.815 | 95 (23.8) | 305 (76.2) | 1.013; 0.634–1.621; *p* = 0.956 | 173 (43.3) | 227 (56.8) | 1.690; 1.120–2.549; *p* = 0.012 |
| **Country income level** | | | | 4.076; 0.605–24.471; *p* = 0.149 | | | | | | |
| LIC | 5 (0.9) | 3 (60.0) | 2 (40.0) | 0.336; 0.180–0.625; *p* = 0.001 | 4 (80.0) | 1 (20.0) | 8.023; 0.827–77.868; *p* = 0.073 | 4 (80.0) | 1 (20.0) | 8.766; 0.908–84.643; *p* = 0.061 |
| LMIC | 361 (66.0) | 46 (12.7) | 315 (87.3) | | 58 (16.1) | 303 (83.9) | 0.323; 0.181–0.577; *p* < 0.001 | 152 (42.1) | 209 (57.9) | 1.230; 0.719–2.105; *p* = 0.450 |
| UMIC | 109 (19.9) | 29 (26.6) | 80 (73.4) | 0.831; 0.419–1.652; *p* = 0.598 | 43 (39.4) | 66 (60.6) | 1.133; 0.602–2.131; *p* = 0.699 | 36 (33.0) | 73 (67.0) | 0.813; 0.430–1.538; *p* = 0.524 |
| HIC | 72 (13.2) | 20 (27.8) | 52 (72.2) | 1.000 | 25 (34.7) | 47 (65.3) | 1.000 | 27 (37.5) | 45 (62.5) | 1.000 |
| **SAR-CoV-2 infection** | | | | | | | 0.689; 0.271–1.751; *p* = 0.434 | | | 0.991; 0.433–2.269; *p* = 0.983 |
| Yes | 33 (6.0) | 3 (9.1) | 30 (90.9) | 0.365; 0.098–1.356; *p* = 0.132 | 9 (27.3) | 24 (72.7) | | 12 (36.4) | 21 (63.6) | |
| No | 514 (94.0) | 95 (18.5) | 419 (81.5) | 1.000 | 121 (23.5) | 380 (76.5) | 1.000 | 207 (40.3) | 307 (59.7) | 1.000 |
| **Know someone who died of COVID-19** | | | | | | | | | | |
| Yes | 219 (40.0) | 36 (16.4) | 183 (83.6) | 0.937; 0.564–1.558; *p* = 0.802 | 53 (24.2) | 166 (75.8) | 0.923; 0.581–1.466; *p* = 0.735 | 73 (33.3) | 146 (66.7) | 0.531; 0.356–0.791; *p* = 0.002 |
| No | 328 (60.0) | 62 (18.9) | 266 (81.1) | 1.000 | 77 (23.5) | 251 (76.5) | 1.000 | 146 (44.5) | 182 (55.5) | 1.000 |
| **Had a close friend and family members who tested positive for COVID-19** | | | | | | | | | | |
| Yes | 178 (32.5) | 26 (14.6) | 152 (85.4) | 0.673; 0.390–1.158; *p* = 0.153 | 45 (25.3) | 133 (74.7) | 1.002; 0.622–1.616; *p* = 0.992 | 69 (38.8) | 109 (61.2) | 1.225; 0.812–1.847; *p* = 0.334 |
| No | 369 (67.5) | 72 (19.5) | 297 (80.5) | 1.000 | 85 (23.0) | 284 (77.0) | 1.000 | 150 (40.7) | 219 (59.3) | 1.000 |
| **Had COVID-19 symptoms but did not get tested** | | | | | | | | | | |
| Yes | 64 (11.7) | 10 (15.6) | 54 (84.4) | 1.271; 0.579–2.786; *p* = 0.550 | 22 (34.4) | 42 (65.6) | 2.473; 1.286–4.755; *p* = 0.007 | 26 (40.6) | 38 (59.4) | 1.205; 0.660–2.198; *p* = 0.544 |
| No | 483 (88.3) | 88 (18.2) | 395 (81.8) | 1.000 | 108 (22.4) | 375 (77.6) | 1.000 | 193 (40.0) | 290 (60.0) | 1.000 |

## 4. Discussion

Findings from the present study indicates that one fifth of adolescents aged 18–19 years experienced depression, a quarter experienced anxiety, four in every ten experienced post-traumatic stress symptoms, and 1 in 10 had SARS-CoV-2 infection during the pandemic. Knowing someone who died from COVID-19 appeared to be associated with lower likelihood of experiencing post-traumatic stress symptoms while having COVID-19 symptoms, but not getting tested appeared to be associated with a higher likelihood of experiencing anxiety symptoms.

One of the strengths of this study is the multicountry-derived sample of adolescents. We also focused on the mental health of adolescents who were transitioning into adulthood, a period characterized by challenges that increase the risk of mental health difficulties. By early adulthood, about half of the population may have already experienced at least one mental health disorder [41–43]. Identifying risk for mental health disorders amongst this age group allows for prompt intervention amongst this population by limiting them to a brief period, thereby reducing the risk of further challenges later in adulthood [44].

The study has a few limitations. First, a nonprobability sample of adolescents was used since they were recruited through an online survey conducted in a limited number of languages. This survey strategy inadvertently excluded populations who could not take the survey due to language restrictions or had poor access to the internet or smart phones. The sample recruited may also have been skewed to those with higher socioeconomic status [45] who were less likely to have mental health problems [46]. In addition, the study collected data on anxiety and depression using single question measures. Though single questions perform well at excluding individuals with mental health challenges among those with no mental health challenges, it performs poorly at confirming mental health problems [47]. In addition, the self-reporting of mental health problems may lead to under-reporting because of the stigma associated with mental health problems [48]. These two factors may lead to the under-reporting of anxiety and depression. Furthermore, this was a cross-sectional study, and conclusions on causal inferences cannot be assumed. Despite these possible limitations, the study highlights a few important findings.

We observed that SARS-CoV-2 infection and other COVID-19-related statuses do not appear to confer a differential risk for mental health disorders among adolescents 18–19 years old [7]. Although a prior study conducted in China indicated that anxiety was almost twice as high for those with compared to those without SARS-CoV-2 infection [49], this multi-country study found no difference in mental health status based on SARS-CoV-2 infection status. The risk of mental health challenges increases with the severity of COVID-19 [50]. The study finding may suggest that the mental health status of adolescents 18–19 with mild COVID-19 may not differ significantly from that of their peers with no infection. Future studies may want to explore the possible differences in the mental health status of adolescents based on severity of infection.

In addition, we observed that having family members who tested positive for COVID-19 was not significantly associated with higher risk of experiencing depression, anxiety, or post-traumatic stress symptoms despite prior evidence to suggest otherwise [24–26]. Additionally, knowing someone who died from COVID-19 was associated with lower risk of post-traumatic stress symptoms in the current study in contrast with prior evidence [51–54]. A prior study had indicated that younger age is associated with post COVID-19 traumatic growth [55]. Post-traumatic growth improves self-efficacy and the ability to deal optimistically with the pandemic [56,57]. It is a positive modification that results from confronting stressful and potentially traumatic events, and a process of finding a sense of personal growth after enduring a psychological struggle [58]. COVID-19, similar to other traumatic events, can be a growth-enhancing experience [59]. It is possible that COVID-19 may be a growth-enhancing experience for adolescents who grieve from the death of people known to them. This postulation needs to be studied further and may be used to enhance adolescents' response to stress emanating from grieving from COVID-19.

Finally, we observed that adolescents who have symptoms of COVID-19 but did not get tested were more likely to experience anxiety. A prior study had indicated that having symptoms of COVID-19 and not getting tested was associated with depression [26]. The observed increased risk for anxiety among adolescents who have symptoms of COVID-19 may be connected with the concern they have about their risk for infection and the risk they pose to others around them. Anxiety can lead to the feeling of uncertainty and fear and can be disruptive to everyday activities [60]. There are multiple reasons that can delay access to COVID-19 tests. These include adolescents not being able to access testing sites or because of concern about stigma; stigma has been shown to keep people away from testing [61]. During pandemics, promoting unrestricted access to low-cost COVID-19 self-testing may help reduce the risk for anxiety associated with not getting to test for COVID-19 [62]. This may be similar to the successes recorded with the increasing uptake of HIV testing with improved access to HIV self-test kits [63,64]. There is the need for more studies to validate this postulation.

The results of the current study do not minimize the risk of mental health challenges adolescents ages 18–19 faced during the COVID-19 pandemic. What the results of this study seem to imply is special considerations should be made for adolescents ages 19–18 when planning for the mental health response of different populations during pandemics, like COVID-19. These individuals are at a life stage when they are transiting from a protective environment to one with less parental control and more independent social involvement. While the risk for post-traumatic stress symptoms seems to be significantly lower for those who knew someone who died from COVID-19, the proportion of those adolescents in the current global study who experienced mental health challenges is high and is justification for instituting mental health support programs for adolescents during the COVID-19 pandemic.

## 5. Conclusions

The present study observed that adolescents aged 18–19 years have similar COVID-19 related mental health risk profiles except for the observed lower risk of post-traumatic stress symptoms associated with knowing someone who died from COVID-19. The current study does not rule out if this age cohort has a COVID-19-related mental health risk profile that differs from that of adults or that of children and younger adolescents. In addition, the study did not identify if the mental health status of adolescents differs by the severity of COVID-19.

**Author Contributions:** Conceptualization, M.O.F.; methodology, M.O.F., A.L.N. and M.E.T.; validation, M.O.F. and A.L.N.; formal analysis, R.A.A.Z.; data curation, M.O.F., R.A.A.Z., M.F.A.Q., J.L., B.G., P.E., A.L.N. and M.E.T.; writing—original draft preparation, M.O.F.; writing—review and editing, M.O.F., R.A.A.Z., M.F.A.Q., J.L., B.G., P.E., A.L.N. and M.E.T.; supervision, M.O.F.; project administration, M.O.F., A.L.N. and M.E.T. All authors have read and agreed to the published version of the manuscript.

**Funding:** This research received no external funding.

**Institutional Review Board Statement:** The study was conducted in accordance with the Declaration of Helsinki. Ethical approval for the study was obtained from the Human Research Ethics Committee at the Institute of Public Health of the Obafemi Awolowo University Ile-Ife, Nigeria (HREC No: IPHOAU/12/1557), Brazil (CAAE N° 38423820.2.0000.0010), India (D-1791-uz and D-1790-uz), Saudi Arabia (CODJU-2006F), and United Kingdom (13283/10570).

**Informed Consent Statement:** Was obtained from all participants involved in the study.

**Data Availability Statement:** Available upon request.

**Acknowledgments:** The authors wish to thank all those who participated in this study.

**Conflicts of Interest:** The authors declare no conflict of interest.

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
