# Peer review of "Associations between Mental Health and COVID-19 Status among 18- and 19-Year-Old Adolescents: A Multi-Country Study"

_adolescents, doi:10.3390/adolescents3010010_

Round 1

Reviewer 1 Report

The study is well executed and reported. The topic is very relevant in the study of Covid-19 in Adolescents. The research problem is properly formulated, and the aims are stated clearly. The research design is applicable, and the statistical methods are explained well. The conclusion is clearly stated.

If I have to say something negative then it is the fact that  about 18 of the 63 references are older than 5 years. The authors should try to use more current references where possible.

Author Response

Thanks for the feedback. This has been extremely helpful in improving the quality of the manuscript. Below are the point-by-point response to the issues raised.

The study is well executed and reported. The topic is very relevant in the study of Covid-19 in Adolescents. The research problem is properly formulated, and the aims are stated clearly. The research design is applicable, and the statistical methods are explained well. The conclusion is clearly stated.

Response: Thanks for the feedback

If I have to say something negative then it is the fact that about 18 of the 63 references are older than 5 years. The authors should try to use more current references where possible.

Response: Thanks for the feedback. We changed reference 22. The majority of the old references are references of the instruments used for measures of study phenomena.

Reviewer 2 Report

Thank you for the opportunity to review the article entitled "Associations between mental health and COVID-19 status among 18- and 19-year-old adolescents: a multi-country study".

This article has merits: the fact that it is a multi-country study, the sample size and the issue of COVID-19. However, I have some comments.

The Introduction is well-written. 

Methods: Why were the responses dichotomized in the Anxiety and Depression measure?

Results: The table is very dense and difficult to follow. I think that the authors should provide at least one figure representation of the results. 

Discussion: The discussion section is sparse. It seems that the authors did not develop the discussion and did not explain their findings, they focused on the strengths and limitations. It also seems that the structure of the discussion section is reversed, the main findings should have been discussed and developed from the beginning. So, I think that there is room to revise this section.

Overall, the article has many advantages but the findings are not new, it can add to the knowledge that already exists and strengthen what is already known. Therefore there is room to refine what the contribution of the article is, and what it adds to the knowledge that already exists. Maybe there was a place to look at differences between countries and see if there are the same results if culture has an influence.  

Author Response

Thanks for the feedback. This has been extremely helpful in improving the quality of the manuscript. Below are the point-by-point response to the issues raised.

Thank you for the opportunity to review the article entitled "Associations between mental health and COVID-19 status among 18- and 19-year-old adolescents: a multi-country study". This article has merits: the fact that it is a multi-country study, the sample size and the issue of COVID-19. However, I have some comments.

The Introduction is well-written. 

Response: Thanks for the feedback

Methods: Why were the responses dichotomized in the Anxiety and Depression measure?

Response: for this study, anxiety and depression are the dependent variables. It was important to report on the proportion of respondents with these experiences for this study.

Results: The table is very dense and difficult to follow. I think that the authors should provide at least one figure representation of the results. 

Response: Thanks a million for the suggestion. The study only developed a table. We feel strongly that developing a figure will require we repeat the reporting of variables. We feel comfortable representing the data as a table. We have therefore, left the table.

Discussion: The discussion section is sparse. It seems that the authors did not develop the discussion and did not explain their findings, they focused on the strengths and limitations. It also seems that the structure of the discussion section is reversed, the main findings should have been discussed and developed from the beginning. So, I think that there is room to revise this section.

Response: Thanks for the observation. We have left the discussion on the strengths and limitations of the study early in line with the requirement of the STROBE guidelines. We have strengthened the discussion by including further discussion of the study findings. We now have a new last paragraph in the discussion section. We wrote: The results of the current study do not minimalize the risk of mental health challenges adolescents 18-19-year-old face during the COVID-19 pandemic. What the results of this study seem to infer is that adolescents 19-18-years-old may be treated as a homogenous entity when planning for the mental health response for different populations during pandemics like the COVID-19. These individuals are at a life-stage when they are transiting from a protective environment to one with less parental control and more independent social involvement. While the risk for post-traumatic stress symptoms seems to be significantly lower for those who knew someone who died from COVID-19, the proportion of those adolescents in the current global study who experienced mental health challenges is high and a reason to institute mental health support programs for adolescents during the COVID-19 pandemic.

Overall, the article has many advantages but the findings are not new, it can add to the knowledge that already exists and strengthen what is already known. Therefore there is room to refine what the contribution of the article is, and what it adds to the knowledge that already exists. Maybe there was a place to look at differences between countries and see if there are the same results if culture has an influence.  

Response: Thanks for the feedback, For this study, we have treated countries as a confounding factor and can therefore not handle It as an independent variable. We think we have made new contributions to population focused studies in the literature. We think that the study results suggests that though adolescents are not homogenous populations, adolescents 18 and 19 years old may be treated as homogenous when managing mental health issues as we found no differences in their mental health profile during this COVID-19 pandemic. No prior study had reported this.

Reviewer 3 Report

The paper needs to improve the data source. 

Author Response

Thanks for the feedback. This has been extremely helpful in improving the quality of the manuscript. Below are the point-by-point response to the issues raised.

The paper needs to improve the data source. 

Response: Thanks for the comments. We have tried to improve the quality of the manuscript. We do not understand the comments. We hope the review conducted may have address the concern of the reviewer.

Reviewer 4 Report

The paper is well written and adds to a list of publications that have attempted to map the impact of the pandemic on mental health. A strength of the work is that it investigates an important segment of the population, i.e. adolescents between 18 and 19 years of age, who received the impact of the pandemic at a developmentally sensitive time in their lives. A limitation of the study, in addition to those already listed by the authors, that makes the results more difficult to interpret is that the data were collected with a self-report questionnaire and this, despite all precautions, may have affected the results. It is recommended that this limitation be reported as well. 

  1. The authors investigated anxiety, depression and PTSD symptoms through two different modalities. While the scale for PTSD is likert type (on 5 items) those for the assessment of anxiety and depressive symptoms are dichotomous type (thus not dimensional). This may certainly have affected the results. This should be highlighted in the limitations and discussed in the results.
  2. In lines 195-198 the authors state that “The only COVID-19 status associated with mental health disorder was knowing someone who died from COVID-19. However, knowing someone who had died from COVID-19 appeared to be associated with lower likelihood of experiencing post-traumatic stress symptoms”. However, in the "Results" section (lines 182-183) they state that “Knowing someone who died from COVID-19 was associated with significantly lower odds of having post-traumatic stress symptoms”. These two statements seem to contradict each other.
  3. Also, again in lines 195-198, the statement "The only COVID-19 status associated with mental health disorder" should be modified. The authors did not make diagnoses but only assessed, through self-report questionnaires, reported symptoms, so it is possible to say that COVID-19 status may be associated with reported symptoms not with actual mental disorders.

Author Response

Thanks for the feedback. This has been extremely helpful in improving the quality of the manuscript. Below are the point-by-point response to the issues raised

The paper is well written and adds to a list of publications that have attempted to map the impact of the pandemic on mental health. A strength of the work is that it investigates an important segment of the population, i.e. adolescents between 18 and 19 years of age, who received the impact of the pandemic at a developmentally sensitive time in their lives. A limitation of the study, in addition to those already listed by the authors, that makes the results more difficult to interpret is that the data were collected with a self-report questionnaire and this, despite all precautions, may have affected the results. It is recommended that this limitation be reported as well. 
Response: we thank the reviewer for the comment. We have included the suggested limitation in the limitation section of the manuscript. We wrote: In addition, the study collected data on anxiety and depression using single question measures. Though single questions perform well at excluding individuals with mental health challenges among those with no mental health challenges, it performs poorly at confirming mental health problems [47]. in addition, the self-reporting of mental health problems may lead to under-reporting because of the stigma associated with mental health problems [48]. These two factors may lead to the under-reporting of anxiety and depression.

The authors investigated anxiety, depression and PTSD symptoms through two different modalities. While the scale for PTSD is likert type (on 5 items) those for the assessment of anxiety and depressive symptoms are dichotomous type (thus not dimensional). This may certainly have affected the results. This should be highlighted in the limitations and discussed in the results.

Response: Thanks once again for raising this valid point. We have addressed this concern by nothing the limitation with the use of a dichotomous measure as indicated above

In lines 195-198 the authors state that “The only COVID-19 status associated with mental health disorder was knowing someone who died from COVID-19. However, knowing someone who had died from COVID-19 appeared to be associated with lower likelihood of experiencing post-traumatic stress symptoms”. However, in the "Results" section (lines 182-183) they state that “Knowing someone who died from COVID-19 was associated with significantly lower odds of having post-traumatic stress symptoms”. These two statements seem to contradict each other.

Response: Thanks for the comments. We studied the two comments and felt the comments were not contradicting. We have edited the sentence slightly to ensure we use the work experiencing more consistently

Also, again in lines 195-198, the statement "The only COVID-19 status associated with mental health disorder" should be modified. The authors did not make diagnoses but only assessed, through self-report questionnaires, reported symptoms, so it is possible to say that COVID-19 status may be associated with reported symptoms not with actual mental disorders.

Response: Thanks for this suggested edit. This statement was deleted.

Reviewer 5 Report

The manuscript is aimed at better understanding the impact that COVID-19 had on the mental health of individuals aged 18-19 years, specifically exploring associations between COVID-19 status and mental health-related variables such as depression, anxiety, and post-traumatic stress.

Overall, the manuscript is well-written and well-articulated. It is of great interest to the Journal and it was a pleasure reading it.

I just notice some minor issues below:

- Lines 125-127: something is wrong with this sentence.

- Lines 136-137: “The questions were adapted from the Pandemic Stress Index”. In what ways? Was the scale translated, reduced, or what else? Please provide more detail on this issue.

- At line 199 the authors begin the discussion of the strengths and limitations of their study. I would rather put this section at the end of the Discussion section, in order to be more fluent in discussing the results.

- Line 218: “first”, “second”, etc. lack an introductory sentence to allow the reader to better understand the subsequent lines.

Author Response

Thanks for the feedback. This has been extremely helpful in improving the quality of the manuscript. Below are the point-by-point response to the issues raised

The manuscript is aimed at better understanding the impact that COVID-19 had on the mental health of individuals aged 18-19 years, specifically exploring associations between COVID-19 status and mental health-related variables such as depression, anxiety, and post-traumatic stress. Overall, the manuscript is well-written and well-articulated. It is of great interest to the Journal and it was a pleasure reading it.

 Response: Thanks for the positive comments

I just notice some minor issues below:

- Lines 125-127: something is wrong with this sentence.

Response: Thanks for the observation. We have completed the sentence. We wrote: Survey responses completed below 7 minutes (the lower limit of the time range to answer the questionnaire during the piloting of the instrument) were excluded from the data analysis

- Lines 136-137: “The questions were adapted from the Pandemic Stress Index”. In what ways? Was the scale translated, reduced, or what else? Please provide more detail on this issue.

Response: Thanks for the observation. We actually adopted rather than adapted as written in the statement. We have corrected this error.

- At line 199 the authors begin the discussion of the strengths and limitations of their study. I would rather put this section at the end of the Discussion section, in order to be more fluent in discussing the results.

Response: We have maintained the position of this statement in line with the STROBE guidelines. The guidelines promote the need for readers to understand the bias about the study and reflect on the findings understanding these biases. We agree with this position of the STROBE guidelines and have therefore retained the position.

- Line 218: “first”, “second”, etc. lack an introductory sentence to allow the reader to better understand the subsequent lines.

Response: Thanks for raising this. We have edited the words and used alternative phrases.

Round 2

Reviewer 2 Report

I thank the authors for the revision..

The authors gave sufficient responses..